# Child Marriage in South Asia: A Systematic Review

**DOI:** 10.3390/ijerph192215138

**Published:** 2022-11-17

**Authors:** S Daarwin Subramanee, Kingsley Agho, Josyula Lakshmi, Md. Nazmul Huda, Rohina Joshi, Blessing Akombi-Inyang

**Affiliations:** 1School of Health Science, Western Sydney University, Campbelltown, NSW 2560, Australia; 2The George Institute for Global Health, Hyderabad 500 082, India; 3School of Population Health, University of New South Wales, Sydney, NSW 2052, Australia; 4Prasanna School of Public Health, Manipal Academy of Higher Education, Manipal 576104, India; 5ARCED Foundation, Dhaka 1216, Bangladesh; 6Translational Health Research Institute, Western Sydney University, Campbeltown, NSW 2560, Australia

**Keywords:** child marriage, teenage marriage, adolescent marriage, educational attainment, mass media exposure, patriarchy

## Abstract

**Background**: Child marriage is a serious public health issue with dire implications at the individual and societal level. Almost half of all child marriages globally originate from South Asia. The aim of this study is to identify consistent factors associated with and resulting from child marriage in South Asia through a review of available evidence. **Methods:** This systematic review adhered to the 2015 Preferred Reporting Items for Systematic review and Meta-Analysis (PRISMA) guidelines. Six computerized bibliographic databases, namely PsycINFO, CINAHL, EMBASE, Ovid Medline, PUBMED, and Scopus were searched. Retrieved studies were exported to EndNote and screened for eligibility using pre-determined criteria. The quality of the included studies was rated using 14 quality appraisal criteria derived from the National Institutes of Health (NIH) Tool. **Results:** A total of 520 articles were retrieved from six databases. Of these, 13 articles met the eligibility criteria and were included in this study. Factors consistently associated with child marriage in South Asia were rural residence, low level of education, poor economic background, low exposure to mass media and religion (Hindu and Muslim in particular countries). Maternal health care factors resulting from child marriage included: low utilization of antenatal care services, low institutional delivery, and low delivery assistance by a skilled birth attendant. **Conclusions:** Child marriage results from an interplay of economic and social forces. Therefore, to address the complex nature of child marriage, efforts targeting improvement in education, employment, exposure to health information via mass media, and gender egalitarianism are required. This systematic review was registered with PROSPERO [CRD42020190410].

## 1. Background

Child marriage, also referred to as early marriage, is a serious human rights violation and detrimental practice that exposes children to increased risk of violence, exploitation, and abuse. It requires serious deliberation and action. Child marriage refers to formal marriages and informal unions in which one or both parties are under 18 years of age and live with a partner as if married [1,2]. Child marriage affects girls and boys, but disproportionately and negatively affects girls, who are more likely to be married off earlier in life than boys, especially in South Asia and Africa. This forced alliance is rooted in harmful pre-existing traditional norms and practices passed down through generations with debilitating effects on the girl child, their families, and society at large [3]. This harmful practice significantly undermines the best interests of the girl child at the individual, societal and national levels [4]. Besides being a human rights issue, child marriage has dire reproductive and sexual health consequences for girls, impeding their overall development and wellbeing [4].

Globally, more than 12 million girls marry at a young age (under 18 years) each year, that is, around 21% of young women marry before the age of 18 [1]. About 37% of these child marriages occur in sub-Saharan Africa, and 30% in South Asia [1]. In 2010, nearly 46% of women aged 20–24 years in South Asia reported being married before the age of 18. This translates to about 24.4 million women in the region, with projections that about 130 million girls are likely to be victims of child marriage between 2010 and 2030 [5].

In recent years, South Asia has experienced a significant decline in child marriage, especially among girls under 15 years [1,2]. This decline is driven predominantly by India, where child marriage rate declined from 47% in 2005 to 27% in 2016 [6]. However, despite the observed decline in child marriage in South Asia from 63% in 1985 to 45% in 2010 and 32% to 17% for girls under 15 years of age [1], this practice continues to be widespread and often concentrated in certain geographic regions or among specific cultural groups, necessitating more targeted efforts to protect adolescents from marriage. These targeted efforts must consider the predisposing factors associated with child marriage within the region.

Previous studies conducted in countries outside South Asia have reported some individual-level factors as being associated with child marriage, including place of residence (mostly rural) [7,8,9], low education level [7,8,9], poverty [7,8], culture [3], lack of exposure to mass media [7,8], low decision-making capacity [10,11], and religion [7,12]. Besides these associated factors, several other factors have also been reported to result from child marriage such as increased school dropouts [12], increased marital violence [10,13], increased maternal morbidity and mortality [14,15], increased risk of unintended pregnancies [16], decreased utilization of antenatal care and postnatal care services [17,18], decreased institutional delivery in health care facilities [17,19], and decreased deliveries assisted by skilled birth attendants [4,17,19]. However, no study has collectively and systematically analyzed the most consistent factors across the entire South Asia region to guide region- and country-specific interventions, which could lead to a decline in child marriage within each South Asian country and across the region. Hence, the aim of this study is to identify the consistent factors associated with and resulting from child marriage in South Asia.

## 2. Methods

### 2.1. Outcome Indicator

The main outcome indicator for this review is child marriage in South Asia. Child marriage is defined as the union or marriage of children or adolescents where one or both individuals are below 18 years of age.

### 2.2. Search Strategy

This systematic review was conducted in accordance with the 2020 Preferred Reporting Items for Systematic reviews and Meta-Analysis (PRISMA) guidelines (Appendix A) [20]. A list of keywords and MeSH terms were generated and used to comprehensively search six computerized bibliographic databases including APA Psych info, CINAHL, EMBASE, Ovid Medline, PUBMED, and Scopus. The search covered studies published between January 2000 and July 2022 in 8 countries of South Asia, namely India, Nepal, Bangladesh, Pakistan, Sri Lanka, Afghanistan, Maldives, and Bhutan. The year 2000 was chosen as the baseline to capture the Millennium Development Goals (MDGs) timeline which ended in 2015, as well as the SDGs which commenced in 2015. All articles retrieved from the respective databases were exported into an EndNote library. To capture relevant additional publications that might have been missed in the initial databases search, a further search of the bibliographical references of eligible studies was conducted, complemented by citation tracking using Google Scholar. The following combination of keywords was used for the search:

(Factor* OR determinant* OR cause* OR correlate* OR predict*)

AND

(Early marriage OR child marriage OR teenage marriage OR adolescent marriage)

AND

(South Asia OR India OR Nepal OR Bangladesh OR Pakistan OR Sri Lanka OR Afghanistan OR Maldives OR Bhutan)

This systematic review was registered with PROSPERO [CRD42020190410].

### 2.3. Inclusion and Exclusion Criteria

Studies were included in this review if they: (i) assessed child marriage where one or both individuals were 18 years or below at marriage; (ii) were conducted in South Asia only; (iii) analyzed factors associated with child marriage; (iv) analyzed factors resulting from child marriage; (v) were published between January 2000 and July 2022; (v) were observational studies (books, policy briefs, case studies, qualitative studies, thesis, or systematic reviews were excluded); (vi) were published in a peer-reviewed journal (commentaries, reviews, and non-peer reviewed research were excluded); (vii) were written in English.

### 2.4. Data Extraction

Retrieved articles from the six databases were exported into an EndNote library, and all duplicates were removed. Articles were then screened based on their titles and abstracts. Following this, the full texts of the retrieved articles were read for relevance, and the articles which met the inclusion criteria were retained. Data extraction and appraisal were independently conducted by two reviewers, and all disagreements between the reviewers were resolved through discussion and consensus. The summary of the selected studies was recorded including author, year of publication, country, number of children and/or adolescents, age of children and/or adolescents, study design and data analysis, study objectives, factors associated with child marriage, factors resulting from child marriage, and quality assessment score.

### 2.5. Quality Assessment

The quality of all eligible studies included in this review was assessed using the National Institutes of Health (NIH) checklist. The checklist measures 14 unique criteria to assess the internal validity of the studies. The reviewed studies were assigned quality scores within the range of 0–14 points [0 if none of the criteria was met, and 14 points if all criteria were met]. Studies were rated as poor quality (score ≤ 5), fair quality (6–9), and good quality (≥10) as shown in Appendix A.

## 3. Results

A total of 520 articles were retrieved from the six databases. After applying the eligibility criteria, 13 studies were retained for this review as shown in Figure 1.

### 3.1. Characteristics of Included Studies

Table 1 shows the summary of selected studies for this review. All retrieved studies were cross-sectional. Studies from only four of the eight countries in South Asia were eligible for inclusion in this review: India, Bangladesh, Nepal, and Pakistan. The number of participants ranged from 155 to 69,751 adolescents. The 14 criteria used to assess the quality of included studies showed that 6 studies were of good quality and 7 studies were of fair quality. The details of the domain-specific score are provided in Appendix A.

### 3.2. Evidence from Reviewed Studies

Table 1 shows the summary of the selected studies. Socio-demographic factors associated with child marriage are rural residency, low level of education, low SES, religion (Hindu and Muslim), female gender, disadvantaged ethnic group, lower caste, lowland ecological zone, low skilled job or unemployed status, low autonomy, and low exposure to mass media. Maternal health care consequences following child marriage include fewer antenatal care (ANC) visits than recommended, no postnatal care (PNC), low delivery assistance by a skilled birth attendant and minimal health facility delivery.

#### 3.2.1. Child Marriage and Socio-Demographic Factors

Studies conducted in Bangladesh [23], Nepal [25] and India [26] reported an association between no/low exposure to mass media and child marriage. In India [21,26], Bangladesh [23,27,28,33], and Nepal [25,30], children from low SES households were found to be more susceptible to child marriage. Studies conducted in Bangladesh [23,27,29], India [26], and Pakistan [31] reported an association between rural residence and child marriage. In contrast, a study conducted in Nepal [25] found that child marriage was prevalent among children in urban areas. Studies conducted in Nepal [22,30], Bangladesh [23,28,29,34,35], India [21,26], and Pakistan [31] reported low educational attainment among most children, their parents and husbands exposed to child marriage. On the other hand, a study from Nepal [25] showed that early married women had husbands with a secondary or higher education level, and two studies conducted in Nepal [25] and Bangladesh [27] reported child marriage among women with a secondary or higher level of education.

Five studies from Nepal [25,30] and Bangladesh [27,29,34] reported child marriage as prevalent among Hindus and Muslims.

#### 3.2.2. Child Marriage and Maternal Health Service

Studies carried out in Bangladesh [23,32], India [24,32], Pakistan [31,32], and Nepal [32] reported that the number of ANC visits attended by early married women were less than four, which is the minimum recommended number of visits. Contrary to this, a study conducted in Nepal [25] reported that a higher percentage of early married women had four or more ANC visits. Most studies in India [24,32,33], Bangladesh [23,32], Nepal [25,32], and Pakistan [31,32] found that early married mothers reported few or no health facility-based deliveries. Studies conducted in Bangladesh [23,32], India [24,32], and Pakistan [31,32] reported low levels of delivery assistance by skilled birth attendants. On the other hand, a study conducted in Nepal [25] showed that delivery for early married mothers was mostly conducted by skilled birth attendants.

## 4. Discussion

This systematic review identified socio-demographic factors associated with, and maternal health care factors resulting from, child marriage in South Asia. Of the socio-demographic factors, the most reported were rural residence, low level of education, poor economic background, religion (Hindu and Muslim in particular countries), and low or no exposure to mass media. On the other hand, reports on maternal health care factors following child marriage showed low utilization of ANC services, low health facility delivery, and the absence of skilled birth attendants during delivery.

In countries where child marriage is prevalent, it is also reported that the practice is not equally distributed, being geographically clustered and more pervasive in rural areas [28,34]. In rural settlements, child marriage is mostly seen among girls with low educational attainment and from poor households, which predisposes them to early childbearing and less access to the health system [30]. The interplay between place of residence, socio-economic status, educational attainment, and uptake of health services as it relates to child marriage is quite complex. Applying the principle of intersectionality [36], it could be inferred that economically disadvantaged girls, with no/low educational attainment, and residing in rural areas are more susceptible to being married off at an early age compared to their more affluent, better-educated counterparts living in urban areas.

Rural residence was reported as being associated with child marriage. This could be associated with limited access to education due to fewer educational facilities in rural areas. Fewer schools restrict the number of children who can obtain proper education. In addition, most rural areas have fewer job opportunities [37], coupled with reduced access to education, which indirectly impacts the economic stability of most families, leading to parents marrying off their daughters at an earlier age for economic security [38]. However, contrary to this finding, a study from Nepal reported that most child marriages occur in urban areas irrespective of the number of educational facilities available [25]. This points to the need for interventions to prevent child marriage in both rural and urban areas.

Religion was associated with child marriage. Every religion has teachings and doctrines which are upheld within the family. In South Asia, the most common religions are Hinduism, Islam, and Buddhism [39]. In this review, Hindu and Muslim religions were reported as being associated with child marriage. However, this is inconclusive, as studies which reported this were conducted in Nepal and Bangladesh which practice predominantly Hindu and Muslim religions respectively. Similarly, a study conducted in the United States of America (USA), a predominantly Christian country, reported the Christian religion to be associated with child marriage [40].

Another consistent factor associated with early marriage is education. The lack of adequate educational opportunities is a significant contributor to child marriage, especially in developing countries. Most often, families facing significant financial hardship resort to marrying off their daughters to relieve some financial burden, instead of paying costly school fees. This practice prevents the girl child from attaining future independence and autonomy through proper education and income-earning jobs. This practice is also prevalent in situations where the parents and the husband have low levels of educational attainment. Education enhances decision-making capacity [41] and has been known to help delay the age of marriage. The education level of parents influences their disposition towards child marriage to a great extent, and their willingness to marry off their young daughters [12]. Parents and husbands with low level of education do not seem to understand the detrimental impact child marriage could have on maternal and neonatal health outcomes. In addition, research has shown that marital violence increases in cases of child marriage where husbands are illiterate [42,43]. In line with our review are studies conducted in Nepal, Zambia, Indonesia, and Bangladesh that showed an inverse association between education and child marriage [7,9,44].

In this review, child marriage was associated with a poor economic background. It has been reported that poverty increases the occurrence of child marriage in South Asia. The poverty situation in South Asia is exacerbated by the extended family model, which involves both immediate family and relatives living together: many parents prefer to marry off their daughters at an early age, and thus be freed of the economic responsibility of their daughters in order to sustain the large families that they need to support [45,46]. Parents of young girls with poor economic backgrounds from countries with a strong culture of acquiring dowry tend to get their daughters married at an earlier age to prevent having to pay a higher dowry at an older age. In certain circumstances, without paying the dowries which increases with increasing age of brides, the young women will not get married, and this is seen as an embarrassment to the family [37].

Another common factor associated with child marriage is a lack of mass media exposure. Most sex education messages are mainly broadcasted via the newspaper, magazines, radio, television, and the internet. Girls with no access to these media channels tend to miss out on these valuable messages. From reviewed studies, it could be extrapolated that families of girls involved in child marriage had minimal exposure to information about the drawbacks of child marriage, the negative impact of marital violence on the girl child, the adverse effects of child marriage on maternal and neonatal health, and other reproductive health relevant information. Consistent with this review, a study on child marriage conducted in Ethiopia also reported an association between low education levels and reduced exposure to mass media among early married women [41].

Most maternal deaths could be prevented if appropriate and timely medical care is provided during pregnancy, childbirth, and the postpartum period. Skilled birth attendance is an essential intervention in improving maternal and newborn outcomes by preventing the occurrence of, and managing, major maternal health complications including eclampsia, obstructed labour, puerperal infection, and obstetric hemorrhage [47]. Skilled birth attendance ensures the availability of health professionals capable of rendering medical assistance during deliveries. This review found low usage of skilled birth attendance following child marriage. Lack of skilled birth attendance was common among uneducated girls who reside in rural areas and had limited access to and knowledge of the importance of having a skilled birth attendant during delivery to prevent or manage severe complications during and after birth [23,31]. In most rural areas in developing countries, almost 43% of deliveries are conducted by traditional birth attendants [48] with limited resources. This makes young women vulnerable to maternal mortality and morbidity due to preventable complications and lack of adequate professional care during and immediately after delivery.

Antenatal care is an important component of standard maternal health care. Comprehensive antenatal care helps prevent and identify prenatal disorders such as pregnancy-induced hypertension, gestational diabetes, anaemia, premature labour, and malaria [49]. This review found that child marriage was associated with a sub-optimal frequency of ANC visits, which negatively impacts maternal and child health. The number of ANC visits made by women is an important predictor of safe delivery and ensuring optimal health for both the mother and the child [50]. The poor attendance at ANC visits among early married mothers observed in this review could result from insufficient knowledge of the importance of ANC in achieving optimal obstetric outcomes. Access to proper ANC is associated with socio-demographic factors such as women’s age, educational attainment, and wealth status [51,52,53].

This review found that child marriage was associated with low utilization of health facilities for delivery. Two studies conducted in Nepal reported that women married before the age of 18 years were less likely to have a health facility based delivery compared to women married after they were over 18 years of age [54,55]. Another study also suggested a large socioeconomic gradient in most developing countries, especially in South Asia, for the use of maternal health care services. Young women with low social standing were reported to have less access to health facility delivery compared to young women with high social standing due to high consultation and service fees posing obstacles to access and utilization [56,57,58].

This systematic review is a comprehensive search of existing literature on the factors associated with and resulting from child marriage in South Asia. Most studies reported were of medium or high quality. Our findings underscore the critical association of several socioeconomic and contextual factors with child marriage, and point to avenues for engagement and intervention to address this important public health, human rights, and equity concern. However, this review also has a few limitations. First, qualitative studies were not included in this review, as eligible studies were restricted to observational studies. The inclusion of qualitative studies in systematic reviews enables the triangulation of study findings, or offers alternative explanations [59]. Second, intervention studies aimed at child marriage prevention by targeting certain exposures were excluded; such studies could provide evidence on the impact of the elimination of identified factors on the occurrence of child marriage, but this is outside the scope of this review. Third, some studies may have been published in other languages besides English, and consequently missed in this review. Fourth, this review did not report the pooled estimate for the effect of each factor on child marriage across South Asia. This is because there were no studies conducted in some countries in South Asia, and the identified factors from the limited countries were measured differently in each study, thus, reporting an estimate for the pooled effect would misrepresent the impact of the factors on child marriage across South Asia. Finally, there were no eligible studies found from most South Asian countries including Sri Lanka, Bhutan, Afghanistan, and Maldives, hence caution should be taken in generalizing the result of this review to the entire South Asia region.

### Policy Implications

This review draws attention to the most common and consistent factors associated with, and resulting from, child marriage in certain South Asian countries. It prompts public health researchers, policymakers, and governments to explore these factors, and could inform policy direction and actions to reduce child marriage in South Asia. Such policy actions should target female children and their parents, improving access to quality education in rural areas, and providing jobs which enhance the economic capacity of families. In addition, increasing exposure to mass media and improving access to adequate maternal health care services such as ANC, skilled birth delivery, and institutional delivery would also be effective. These strategies will yield a more sustainable reduction in child marriage, thus setting the region on its path to achieving 8 out of 17 SDGs by year 2030, including those related to: poverty, food security, health, education, gender equality, economic growth, and peace and justice [60].

## 5. Conclusions

This systematic review reported several factors associated with and resulting from child marriage across South Asia. The most consistent factors reported were rural residence, religion (Hindu and Muslim in particular countries), low level of education, poor economic background, and low/no exposure to mass media. The common factors resulting from child marriage were reproductive and maternal factors such as reduced utilization of ANC services, fewer institutional deliveries, and the absence of a skilled birth attendant during delivery. Factors such as place of residence, education, and economic background intersected and were associated with higher rates of child marriage. This study serves as a needs assessment indicator to further explore the factors associated with and resulting from child marriage in countries of South Asia with no representation in research on child marriage.

## Figures and Tables

**Figure 1 ijerph-19-15138-f001:**
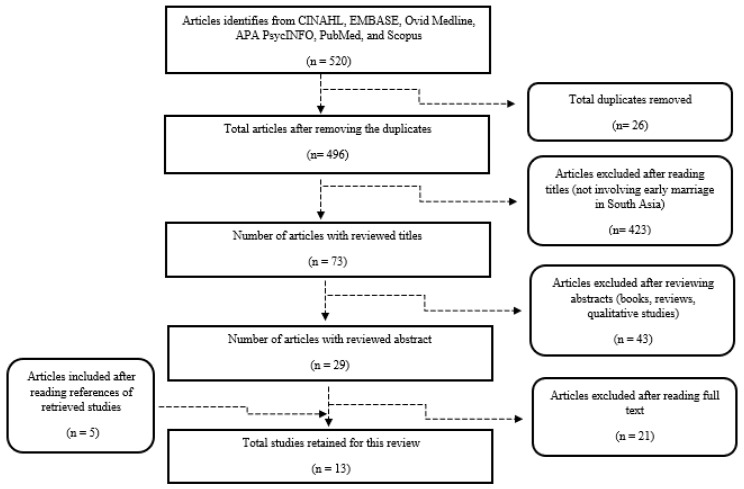
Flow chart for study selection based on PRISMA 2015 guidelines.

**Table 1 ijerph-19-15138-t001:** Summary of selected studies.

Author, Year, Country [Ref]	Number of Participants, Age	Study Design, Data Analysis	Study Objective	Factors Associated with Child Marriage	Factors Resulting from Child Marriage	Quality Assessment Score (0–14)
Binu et al., 2022, India [21]	N = 1459 15–17 years	Cross-sectional study, Logistic regression	To identify the direct and indirect sociodemographic factors associated with child marriage among 15–17 years old girls in India	Low level of education of mother and girl child, low socioeconomic status (SES), caste	N/A	8 Fair
Yogi et al., 2020, Nepal [22]	N = 155 18.4 years	Cross-sectional study, Logistic regression	To examine factors associated with early marriage among the people in rural communities of Nepal	Being female and low level of education	N/A	6 Fair
Uddin et al., 2019, Bangladesh [23]	N = 16099 15–19 years	Cross-sectional study, Logistic regression	To examine the association between child marriage and women’s use of institutional delivery care services	Rural residence, low level of education, low exposure to mass media, low SES	Minimal health facility delivery, less likely to have skilled birth assistance, had fewer number of antenatal care (ANC) visits	11 Good
Paul et al., 2019, India [24]	N = 69751 <18 years	Cross-sectional study, Logistic regression	To examine the association between child marriage and utilization of maternal health care services	N/A	Less likely to have four ANC visits, less likely to have ANC visit within the first trimester, less likely to deliver in a health facility, and less likely to have a skilled birth attendant at delivery	11 Good
Sekine et al., 2019, Nepal [25]	N = 3970 15–19 years	Cross-sectional study, Logistic regression	To assess the total effect of child marriage on the utilization of maternal health services in Nepal	Urban residency, Hindu religion, disadvantaged ethnic group, lowland ecological zone, low SES, highest education being secondary or higher, highest husbands’ education being secondary or higher, unemployed, or low skilled, no exposure to mass media	Regular ANC visits, more deliveries by a skilled birth attendant, fewer deliveries in a health facility, and no postnatal care (PNC) coverage within 24 h of delivery	12 Good
Mehra et al., 2018, India [26]	N = 1770 10–19 years	Cross-sectional study, Logistic regression	To assess the impact of a multi-pronged community-based intervention on early marriage, early pregnancy, and school retention among young people in two states of India	Rural residence, low level of education, no exposure to peer educators, low SES, lower caste, being female, no exposure to mass media, low skilled jobs mostly as housemaids	N/A	9 Fair
Hossain et al., 2016, Bangladesh [27]	N = 11,780 ≤17 years	Cross-sectional study, Logistic regression	To determine the prevalence, and factors associated with, child marriage among Bangladeshi women	Rural residence, Muslim religion, having secondary level of education, low SES, low level of education of husband	N/A	10 Good
Islam et al., 2016, Bangladesh [28]	N = 17,808 12–18 years	Cross-sectional study, Logistic regression	To investigate the regional variations in the prevalence of child marriage in Bangladesh	Low level of education of girl child, and husband, unemployed, low SES	N/A	9 Fair
Kamal et al., 2014, Bangladesh [29]	N = 59,792 <18 years	Cross-sectional study, Logistic regression	To examines the trends and determinants of child marriage in Bangladesh	Low level of education of husband, unemployed or unskilled workers, Muslim religion, rural residence	N/A	10 Good
Sah et al., 2014, Nepal [30]	N = 246 <18 years	Cross-sectional study, Chi-Square test	To find out the factors associated with early age marriages in Dhankuta Municipality	Low level of wife and husband education, low SES, Hindu religion	N/A	6 Fair
Nasrullah et al., 2013, Pakistan [31]	N = 1404 <18 years	Cross-sectional study, Logistic regression	To assess the association between child marriage and maternal health care services use in Pakistan	No formal education, rural residence	Decreased likelihood of ANC, decreased likelihood of delivery assistance by skilled birth attendant, delivery at home	9 Fair
Godha et al., 2013, India, Bangladesh, Nepal, Pakistan [32]	N = 14,628 (India), 2129 (Bangladesh), 1658 (Nepal), 1546 (Pakistan) <18 years	Cross-sectional study, Logistic regression	To assess the association of child marriage with maternal health care use outcomes in four South Asian countries: India, Bangladesh, Nepal, and Pakistan	N/A	Fewer ANC visits, reduced likelihood of delivery in a health care facility, and less likely to have a skilled birth attendant present during delivery	10 Good
Santhya et al., 2010, India [33]	N = 8314 <18 years	Cross-sectional study, Logistic regression	To compare the reproductive characteristics of young women who had married before 18 years of age with those who had married after 18 years of age	N/A	Less likely to have first birth in a health facility	9 Fair
Kamal et al., 2010, Bangladesh [34]	N = 9572	Cross-sectional study, Logistic regression	To examine the regional and contextual disparities of child marriage	Low level of education, rural residence, Muslim religion, and low SES	N/A	9 Fair
Rahman et al., 2005, Bangladesh [35]	N = 3362 10–19 years	Cross-sectional study, Logistic regression	To investigate the factors influencing adolescents’ attitudes towards early marriage	Low education level, low parental education level, low skilled job, lesser dowry, no autonomy	N/A	6 Fair

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
