# Peer review of "Child Marriage in South Asia: A Systematic Review"

_ijerph, 2022, doi:10.3390/ijerph192215138_

Round 1

Reviewer 1 Report

Thank you for addressing an important issue for women's health.

In the title: "early" is too vague a descriptor for the kind of marriages you review. In the body of the paper, you use the term "child marriage." Please use it in the title. That would change reader interest from lukewarm to serious.

In the Discussion section, no notice is given about a major limitation of this paper: a lack of information on morbidity and mortality of the girls after marriage, especially after pregnancy, and a lack of information on morbidity and mortality of their offspring.  Yet the Introduction and Discussion assert (in only one reference each) that such factors have been assessed and that the results are "dire." This would be a vital addition if you can make it.

Author Response

Thank you for addressing an important issue for women's health.

Comment: In the title: "early" is too vague a descriptor for the kind of marriages you review. In the body of the paper, you use the term "child marriage." Please use it in the title. That would change reader interest from lukewarm to serious.

Response: Thank you for your comment. We have changed the term from “early marriage” to child marriage throughout the manuscript.

Comment: In the Discussion section, no notice is given about a major limitation of this paper: a lack of information on morbidity and mortality of the girls after marriage, especially after pregnancy, and a lack of information on morbidity and mortality of their offspring.  Yet the Introduction and Discussion assert (in only one reference each) that such factors have been assessed and that the results are "dire." This would be a vital addition if you can make it.

Response: Thank you for your comment/observation. The aim of our study was to identify the consistent individual level factors associated with and the maternal health service factors resulting from child marriage in South Asia from available literature. Whilst we recognise that these factors could lead to increased morbidity and mortality of the girl child. Assessing the direct health impact of child marriage on the girl child and her offspring is out of scope for our review.

Reviewer 2 Report

This systematic review is a paper that examines early marriage in South Asia, a serious public health problem with dire consequences at the individual and societal levels. The manuscript is well written throughout, but raises several questions.

1. List and define all outcomes for the data sought.

2. List and define all other variables for the data sought (e.g., participant and intervention characteristics, funding source).

3. Describe the estimation used for missing or unclear information. 

4. Describe the methods used to assess the risk of bias in the included studies, including details of the tools used, the number of reviewers who evaluated each study, whether they worked independently, and if applicable, details of any automated tools used in the process.

Author Response

This systematic review is a paper that examines early marriage in South Asia, a serious public health problem with dire consequences at the individual and societal levels. The manuscript is well written throughout but raises several questions.

Response: Thank you for your kind words.

Comment: List and define all outcomes for the data sought.

Response: The main outcome indicator for our study was child marriage and this was defined in the methods section under sub-section “Outcome indicator”.

Comment: List and define all other variables for the data sought (e.g., participant and intervention characteristics, funding source).

Response: The inclusion and exclusion criteria define the necessary data sought for inclusion into our study. In addition, our study received no funding, and this has been clearly stated in the manuscript and communicated to the journal.

Comment: Describe the estimation used for missing or unclear information. 

Response: Our study is a systematic review; therefore, statistical estimation is not applicable. Our study reported the evidence obtained from reviewed studies. As stated in the manuscript, we only selected studies written in English for easy comprehension and any unclear information was discussed between the reviewers.

Comment: Describe the methods used to assess the risk of bias in the included studies, including details of the tools used, the number of reviewers who evaluated each study, whether they worked independently, and if applicable, details of any automated tools used in the process.

Response: This information has already been provided in the manuscript. Please see “Data extraction” and “Quality assessment” sub-sections in the methods section.